# Tracking the clinico-microbiological profile and molecular characterization of dengue cases during the monsoon-season in Belagavi, Karnataka

Rekha Patil[1☯], Asif Kavathekar[2☯], Madhav Prabhu[1☯], Madhavi Patil[2], Ameya Kejriwal[1], Kamran Zaman[2]*, Jyothi Bhat[2]*, Subarna Roy[2]

1 Department of General Medicine, Jawaharlal Nehru Medical College, Belagavi, India, 2 Indian Council of Medical Research - National Institute of Traditional Medicine, Belagavi (ICMR-NITM Belagavi), Belagavi, India

☯ Authors contributed equally and shared first authorship
* bhatdr@gmail.com (JB); kamran3zaman@gmail.com (KZ)

## Abstract

This study aimed to identify and characterise circulating DENV serotypes during the 2024 monsoon season in Belagavi, Karnataka and to analyze the clinical and demographic profiles of the hospitalized dengue cohort and correlate these patterns with the identified serotypes. A prospective, hospital-based observational study was conducted at a tertiary care centre in Belagavi, Karnataka, from June to September 2024. A total of 89 dengue-confirmed cases out of the 794 dengue-positive cases detected using NS1 antigen and IgM antibody by ELISA technique were subjected to serotyping by RT-PCR targeting the CprM gene. A subset of positive RT-PCR specimens was subjected to sequencing and phylogenetic analysis. The clinical details were analysed for demographic features, clinical manifestations, and diagnostic test results to establish correlations with serotype distribution. Of the 89 positive dengue patients, 28 (31.46%) were confirmed positive for dengue virus NS1 antigen by ELISA, while 87 (97.75%) patients tested positive for anti-dengue IgM antibodies. Dengue serotyping of the 28 NS1-positive samples was carried out, of which 23 (82.1%) were positive. DENV-2 (91.3%) was the predominant serotype, followed by DENV-1 and DENV-3, each identified in one case (4.3%, respectively). Genetic analysis of DENV-2 isolates revealed close phylogenetic relationships with the strains from Bhopal, indicating possible regional transmission dynamics. Fever and myalgia were the predominant symptoms across all serotypes, while chills, headache, and retro-orbital pain were associated with DENV-2. The present study is the first comprehensive molecular surveillance report from the northern part of Karnataka demonstrating the predominance of DENV-2 serotype, with distinct clinical and demographic patterns. The high proportion of late-phase diagnoses of dengue underscores the need for improved early detection strategies and timely interventions to manage

which permits unrestricted use, distribution, and reproduction in any medium, provided the original author and source are credited.

**Data availability statement:** All data are in the manuscript and the supporting information files. Raw data is available from S4 Table. The sequence has been submitted to Genbank and the accession numbers are available from S3 Table.

**Funding:** This work was supported by the institutional VRDL laboratory funded by the Department of Health Research (DHR), New Delhi and the Indian Council of Medical Research (ICMR), New Delhi (Grant no: R.15012/03/2023-HR-VRDL to KZ). The funders had no role in the study design, data collection and analysis, the decision to publish, or the preparation of the manuscript.

**Competing interests:** The authors have declared that no competing interests exist.

dengue outbreaks, especially during high-risk transmission periods. These results underline the importance of continuous molecular surveillance to track viral evolution and monitor serotype shift and genetic variations.

---

## Author summary

Dengue fever remains a major, persistent threat to public health across the world, and effective management requires constantly tracking serotypes circulating in different regions. This study provides the first-ever molecular characterization of the Dengue virus causing illness in Belagavi, North Karnataka, an area previously lacking this essential, regional-specific surveillance data.

We analyzed a cohort of 89 hospitalized patients confirmed with Dengue during the 2024 monsoon season. Our molecular findings clearly identified Dengue virus Serotype 2 (DENV-2) as the overwhelmingly predominant strain. Clinically, we observed that most patients sought hospital care in the late stages of their illness, still presented with concerning clinical features, most notably a high frequency of severe thrombocytopenia (low platelet count).

By identifying the current dominant viral strain (DENV-2) and describing the clinical patterns in the most severe cases requiring hospitalization, this research offers crucial, actionable information for local health authorities. This data is vital for guiding public health interventions, such as tailored vector control strategies, and for better training clinical staff in the timely diagnosis and critical management of this endemic neglected tropical disease.

## 1. Introduction

Dengue is the most rapidly spreading vector-borne viral disease in the world and is the second most prevalent mosquito-borne disease in humans after malaria [1]. In India, Dengue has been a persistent public health concern with a rising trend in concurrent infections and the circulation of multiple dengue virus (DENV) serotypes over recent decades. In 2024, data from the National Centre for Vector-Borne Disease Control (NCVBDC) showed an increase in dengue patients in the states of Karnataka, Kerala, and Tamil Nadu [2]. Over the past decades, DENV transmission has expanded beyond urban areas to peri-urban and rural regions of India [3–6].

Dengue infection is caused by four distinct serotypes of the Dengue virus (DENV 1–4), which are transmitted through the bite of infected Aedes mosquitoes, primarily *Aedes aegypti* and *Aedes albopictus*. DENV serotypes exhibit approximately 25–40% amino acid variability, further classified into genotypes based on around 3% amino acid sequence differences [7]. In addition to the four primary serotypes, a fifth serotype (DENV-5) was identified in 2013, following a sylvatic transmission cycle [8].

Despite substantial advances in vaccine development for DENV, an effective tetravalent vaccine capable of neutralizing all four DENV serotypes remains elusive, largely due to antibody-dependent enhancement (ADE) and serotype-specific immune responses [9,10].

These challenges of increasing prevalence and lack of an effective vaccine underscore the critical importance of comprehensive molecular surveillance to track the circulating serotypes, monitor evolutionary trajectories, and identify the key mutations that may play a vital role in impacting the diagnostic accuracy, vaccine efficacy, and the disease severity in terms of patient outcomes.

In 2024, a high burden of dengue has been reported in Karnataka with 32886 dengue positive cases and 27 deaths, according to the NVBDCP (2) and there is a critical gap in understanding the circulating serotypes, molecular epidemiology, and genetic diversity of the circulating DENV serotypes. This knowledge deficit hampers effective surveillance and control measures and clinical management strategies. To address this gap, the present study aimed to conduct a comprehensive molecular surveillance and genetic characterization of the circulating DENV serotypes in Belagavi, representing North Karnataka, during the 2024 monsoon season. This study focused on patients hospitalized with dengue in a tertiary care setting in Belagavi. Additionally, this study also analysed the serotype distribution with demographic patterns, clinical manifestation, and diagnostic profile markers to enhance the understanding of the regional dengue epidemiology.

## 2. Ethical statement

The study was conducted in accordance with the institutional requirements. The study was approved by the Indian Council of Medical Research-National Institute of Traditional Medicine, Belagavi (ICMR-NITM Belagavi) Institutional Ethics Committee (IHEC/NITM/IHEC approval/2024-Sep/032 dated 17.09.2024) and the Institutional Ethics Committee of Karnatak Lingayat Education Academy of Higher Education and Research (KAHER) (KAHER/EC/2024–25/D-26082403 dated 26th August 2024). Informed written consent was obtained from the participants/parents/guardians wherever applicable.

## 3. Materials and methods

### 3.1 Study design and setting

A prospective, hospital-based observational study was conducted on patients admitted with dengue infection in KLE's Prabhakar Kore Hospital & Research Centre, a tertiary care setting in Belagavi, Karnataka and the Virus Research and Diagnostic laboratory (VRDL) of Indian Council of Medical Research-National Institute of Traditional Medicine, Belagavi (ICMR-NITM Belagavi), Karnataka, during the monsoon season (June-September 2024). A total of 2089 patients visited the hospital with febrile illness, of which 794 patients had confirmed dengue infection. To understand the circulating serotype, a subset of these patients was included in this study. A total of 89 cases included in our study represent a consecutive subset of these 794 patients who were admitted to the tertiary care centre (hospitalized cohort) and whose residual samples were subsequently forwarded to the ICMR-NITM VRDL lab for advanced molecular and serological confirmation/serotyping. The WHO dengue severity guideline was utilised to classify the cases as dengue with or without warning signs. [11]

### 3.2 Sample collection and processing

Blood samples (3–5 ml) were collected from the dengue-confirmed patients. The healthcare staff collected patients clinical and demographic data at the hospital. The samples were sent to the VRDL laboratory, maintaining the cold chain.

### 3.3 Serological tests

Initial dengue screening at the hospital was performed using the Bioline Dengue Duo (Dengue NS1 Ag + IgG/IgM) RDT (Abbott Diagnostics, South Korea), which detects NS1 antigen and anti-dengue IgG/IgM antibodies. Further confirmation

was carried out at the VRDL Laboratory using the Panbio Dengue early ELISA (Abbott Ltd.) and the Dengue IgM capture ELISA kit (ICMR-NIV, Pune), following the manufacturer's protocol.

### 3.4 Molecular testing

Viral RNA was extracted from 140 µl of serum using a QIAamp Viral RNA extraction kit (Qiagen, Hilden, Germany) according to the manufacturer's instructions. The quality of extracted RNA was determined and was stored at -20°C until further use. All NS1 antigen-positive samples were subjected to dengue serotyping using the DiA Sure Dengue Genotyping Kit (DiaSure Biotech, India). cDNA was synthesised using the Prime Script 1st Strand cDNA synthesis Kit (Takara Bio USA, Inc.), and RT PCR targeting the capsid pre-membrane (CprM) gene was carried out as described by Lanciotti et al [12].

### 3.5 Sequencing and phylogenetic analysis

Amplified PCR products were visualized and the target amplicon was purified using QIAquick Gel Extraction Kit (Qiagen, Hilden, Germany). Bidirectional sequencing was performed by the Sanger method (Barcode Biosciences, Bangalore, India). The obtained sequences were aligned by ClustalW (version 2.1, European Bioinformatics Institute, Hinxton, UK), and a phylogenetic tree was generated using Molecular Evolutionary Genetics Analysis (MEGA) software version 11 using the Maximum Likelihood method (Tamura-Nei model) with 1,000 bootstrap replicates.

### 3.6 Statistical analysis

Quantitative variables were reported as mean ± SD or median (IQR), based on normality assessed by the Kolmogorov–Smirnov or Shapiro–Wilk tests. Group comparisons used the independent samples t-test or Mann–Whitney U test, as appropriate. Statistical significance was set at $p < 0.05$. Analyses were performed using SPSS version 25 (IBM Corp., Armonk, NY). The bubble plot was created using the Python package matplotlib v3.9.2.

## 4. Results

A total of 794 patients were dengue-positive (NS1 and/or IgM) cases identified by routine hospital diagnostics (Bioline Dengue Duo RDT) during the study period. The 89 cases included in our study represent a consecutive subset of these 794 patients who were admitted to the tertiary care centre (hospitalized cohort) and whose residual samples were subsequently forwarded to the ICMR-NITM VRDL lab for advanced molecular and serological confirmation/serotyping.

### 4.1 Demographic characteristics

Demographic details of 89 laboratory-confirmed patients revealed a significant gender disparity, with males (69.66%) more affected than females (30.33%). The median age of the patients was 34 years (range 17–71 years). The age distribution analysis revealed the highest incidence in the 21–30 years group (32.58%), followed by the 31–40 years group (22.47%). The older age groups (≥51 years) accounted for only 10.11% (9/89) of the total patients. The analysis of the monthly distribution of dengue patients revealed, July recorded the highest incidence with 62 patients (69.66%), followed by August (15.73%).

### 4.2 Dengue diagnostic marker

Initial screening was performed using the Bioline Dengue Duo (Dengue NS1 Ag + IgG/IgM) RDT at the hospital and our subsequent VRDL analysis focused on NS1 and IgM ELISA (Panbio Dengue early ELISA and Dengue IgM capture ELISA kit) as the gold standard markers for acute/recent infection in clinical management. The results of the ELISA tests conducted at the VRDL laboratory were considered. Of the 89 patients evaluated, 28 (31.46%) tested positive for NS1 antigen, while 87 (97.75%) were positive for anti-dengue virus IgM antibodies. A total of 26 patients (29.21%) tested positive

for both NS1 and IgM. Among the NS1-positive patients, 18 (64.28%) were male and 10 (35.71%) were female. Similarly, IgM positivity was observed in 61 males (70.11%) and 26 females (29.89%). Co-positivity for NS1 and IgM was recorded in 17 males (65.38%) and 9 females (34.61%) (**Table 1**).

### 4.3 Clinical presentation and parameters

Fever was the most common symptom, observed in 86 (96.63%) of 89 cases, followed by chills in 59 (66.29%), myalgia in 56 (62.92%), and headache in 30 (33.70%) cases. Other symptoms included abdominal pain in 17 (19.10%), vomiting in 14 (15.736%), and nausea in 12(13.48%) cases. Retro orbital pain and body aches were noted in 7(7.86%) cases. Most cases experienced fever lasting an average of 4 days (range of 1–8 days). The relevant laboratory parameters are shown in the table below (**Table 2**).

As per the WHO dengue severity guidelines, the clinical signs and symptoms were assessed in all 89 patients. The history of fever at the time of admission was identified as the most common symptom among all. Based on dengue warning signs, patients with fever with abdominal pain & tenderness were categorised as dengue with warning signs (DWS), and the rest of the patients were categorised as dengue without warning signs (DWOS). None of the patients belonged to the severe dengue category (**Table 3**).

**Table 1. Demographic characteristics of Dengue positive markers.**

| Variables | NS1 positive, n (%) | IgM positive, n (%) | NS1+IgM positive, n (%) |
|---|---|---|---|
| Overall | **28 (31.46%)** | **87 (97.75%)** | **26 (29.21%)** |
| Sex | | | |
| Male | 18 (64.28%) | 61 (70.11%) | 17 (65.38%) |
| Female | 10 (35.71%) | 26 (29.89%) | 9 (34.61%) |
| Age group | | | |
| 10–20 years | 5(17.86%) | 12(13.79%) | 5(19.23%) |
| 21–30 years | 9(32.14%) | 28(32.18%) | 8(30.77%) |
| 31–40 years | 4(14.28%) | 20(22.99%) | 4(15.38%) |
| 41–50 years | 6(21.42%) | 19(21.84%) | 6(23.07%) |
| 51–60 years | 2(7.14%) | 4(4.60%) | 1(3.85%) |
| 61-70 years | 2(7.14%) | 3(3.45%) | 2(7.69%) |
| 71-80 years | 0 | 1(1.15%) | 0 |

**Table 2. Clinical features and correlation with diagnostic markers NS1 & IgM.**

| Variables | NS1 Positive | | | IgM Positive | | | NS1+IgM Positive | | |
|---|---|---|---|---|---|---|---|---|---|
| | Median | Q1 | Q3 | Median | Q1 | Q3 | Median | Q1 | Q3 |
| Total leucocyte count (10³/μl) | 3.9 | 2.1 | 5.1 | 4.6 | 2.6 | 6.4 | 3.9 | 12.9 | 16.0 |
| DLC neutrophil (%) | 68 | 55 | 76 | 65 | 54 | 73 | 67 | 55 | 76 |
| DLC lymphocyte (%) | 24 | 12 | 36 | 27 | 16.5 | 36 | 24 | 2 | 5.4 |
| Platelet count (μL) | 106000 | 47000 | 162000 | 87000 | 32000 | 150000 | 106000 | 28000 | 163000 |
| SGOT/ AST (U/L) | 78 | 41 | 141 | 85 | 54 | 141 | 82 | 58 | 205 |
| SGPT/ ALT (U/L) | 60 | 36 | 112 | 65.5 | 35 | 107 | 66 | 37 | 126 |
| Creatinine (mg/dl) | 0.9 | 0.8 | 1.1 | 0.9 | 0.7 | 1.1 | 0.9 | 0.8 | 1.1 |
| BUN (mg/dl) | 10.2 | 8.8 | 14.1 | 9.4 | 7.7 | 14.5 | 10.5 | 9.3 | 15.4 |

**Table 3.** Clinical parameters and their significance with dengue severity.

| Variables | Dengue infection (n=89) | | | | | | Mann-Whitney U test P-value |
| --- | --- | --- | --- | --- | --- | --- | --- |
| | Dengue with warning signs (DWS) (n=19) | | | Dengue without warning signs (DWOS) (n=70) | | | |
| | Median | Q1 | Q3 | Median | Q1 | Q3 | |
| PULSE (bpm) | 84 | 78 | 90 | 84.5 | 74 | 96 | 0.7 |
| WBC Count (10³µl) | 3.8 | 2.6 | 6.3 | 4.5 | 2.2 | 7.7 | 0.7 |
| DLC N (%) | 58 | 55 | 74 | 67 | 55 | 73 | 0.6 |
| DLC L (%) | 31 | 16 | 34 | 27 | 17 | 36 | 0.4 |
| Platelet count ((µl)) | 104500 | 34000 | 150000 | 88500 | 23000 | 144000 | 0.8 |
| SGOT/ AST (U/L) | 123.5 | 44 | 133 | 78 | 63 | 201 | 0.1 |
| SGPT/ ALT (U/L) | 66 | 32 | 98 | 63 | 41 | 175 | 0.3 |
| Creatinine (mg/dl) | 1.1 | 0.7 | 1.0 | 0.9 | 0.9 | 1.2 | 0.009* |
| BUN (mg/dl) | 11.0 | 7.2 | 13.5 | 9.1 | 7.7 | 15.7 | 0.2 |
| Days of hospitalisation | 4 | 3 | 6 | 4 | 4 | 6 | 0.5 |

### 4.4 Co-morbid conditions and clinical management

Eight patients were known cases of Type 2 diabetes mellitus, 4 were hypertensive, while 6 individuals were both diabetic and hypertensive. Eighteen patients spent more than a week in the hospital, with an average stay of 5.4 days. Of these, 1 patient had the longest stay of 25 days and was the only mortality case who developed encephalopathy with intracranial bleeding. Of the 89 patients, 88 recovered while one succumbed to the illness. Sixty-seven patients presented with thrombocytopenia (≤150000).

### 4.5 Dengue virus serotype and clinical features distribution

Twenty-eight NS1 positive samples were tested for DENV serotype; 23 (82.14%) were found positive. Among the 23 laboratory-confirmed dengue patients with known serotype data, DENV-2 was the predominant serotype, accounting for 91.3% of the patients, while DENV-1 and DENV-3 were each detected in only a single patient (4.3% each). The clinical presentations observed across these serotypes varied slightly, with DENV-2 exhibiting a broader and more diverse symptom profile.

Fever emerged as the most consistent observed symptom, being present in 100% of DENV-1 and DENV-3 patients, and in 95.23% of DENV-2 patients. This suggests fever as the most characteristic and universal clinical feature of dengue infection across all serotypes. Myalgia was observed in all DENV-1 and DENV-3 patients in our cohort and in one-third of DENV-2 infections. Other clinical features, differences among serotypes are reflected in **Fig 1**.

The hematological analysis revealed thrombocytopenia as an important feature in all serotypes. While Grade I thrombocytopenia (75,000–150,000/µL) was the most frequent finding among genotyped cases (10/23), both DEN2 and the single DEN3 case presented at the critical Grade IV level (<25,000/µL), indicating severe hematological compromise. The serotype 2 cases showed profound liver derangement, with SGOT levels reaching over 1100 U/L and SGPT over 640 U/L (**S2 Table**), indicating significant hepatic involvement. While generally stable, some patients, predominantly DENV-2, showed isolated acute kidney injury markers **(Tables A and B in S1 Table)**.

### 4.6 Phylogenetic analysis

Of the 23 DENV-positive samples detected via real-time RT-PCR, seven (30.4%) yielded amplicons upon conventional RT-PCR. However, high-quality sequence data amenable to phylogenetic inference were obtained from a single sample,

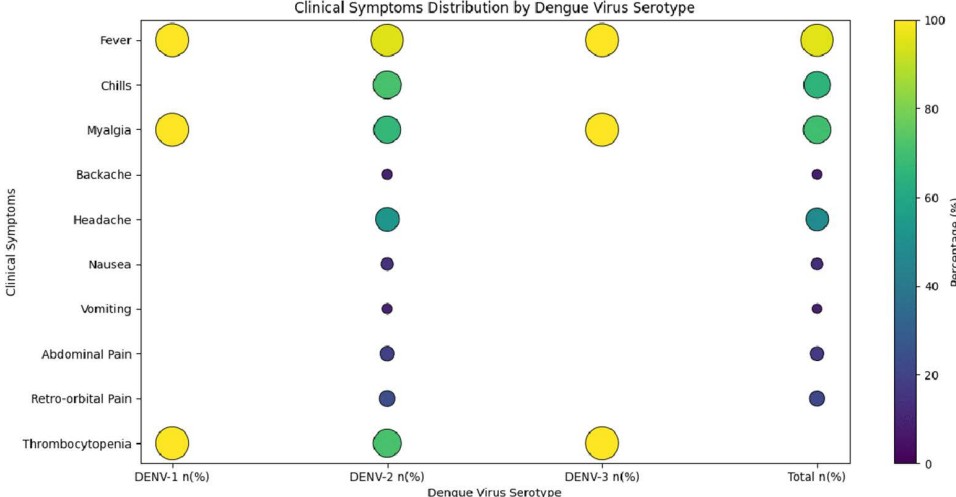

**Fig 1. Bubble plot† illustrating the distribution of clinical symptoms among patients infected with different dengue virus (DENV) serotypes in Karnataka, India (n = 23).** †Each bubble's size corresponds to the number of patients exhibiting that symptom, with colour intensity further reflecting the count.

identified as DENV-2, and submitted to GenBank (Accession No. PV992631). The sequence exhibited 100% identity with reference strain PP082856.1 (DENV-2).

A phylogenetic analysis included reference sequences from multiple regions across India and other countries to determine the evolutionary relationship of the DENV-2 strain detected in Belagavi during the 2024 monsoon season (**Fig 2**). The Belagavi strain (24062 Belagavi 2024) clusters closely with Bhopal, 2023 (OR888735), indicating significant genetic similarity and suggesting they may share a recent common ancestor among circulating strains within central India.

A pairwise distance matrix (**S2 Table**). confirmed minimal divergence (D = 0.002) between the Belagavi strain and other recent Indian isolates, whereas one Bhopal,2023 strain (OQ842500) showed slightly higher divergence (D = 0.004), suggesting a closely related sub-lineage.

## 5.  Discussion

This study provides valuable insights into the molecular surveillance and genetic characterisation of circulating Dengue virus (DENV) serotypes during the 2024 monsoon season in Belagavi, Karnataka. The study involving 89 laboratory-confirmed dengue patients in Belagavi, Karnataka, shows a clear male predominance of 69.7% and a median age of 34 years. The highest number of cases occurs in young adults aged 21–30 years (32.6%). Young adults may be at higher risk because they are of working age and hence might have a higher exposure. These results are consistent with earlier reports indicating higher dengue incidence among males in India, likely due to increased outdoor activities and cultural practices that expose men more to mosquito bites [6,15]. The age distribution, especially the concentration in adolescents and young adults, supports the known trend that clinical dengue risk increases with age, with a significant increase in symptomatic cases after adolescence [16,17]. Seasonal analysis revealed peak transmission in July (69.7% of cases), coinciding with the monsoon-driven surge in Aedes vector populations in different parts of India [18] and a similar upsurge was noted in Karnataka [19].

The low NS1 positivity rate observed during the study period suggests a delay in seeking medical care or in the timing of sample collection, which may have missed early patients. This is a crucial finding and a limitation of our tertiary care setting. The significant co-positivity of NS1 and IgM in some patients further supports the hypothesis that active viral

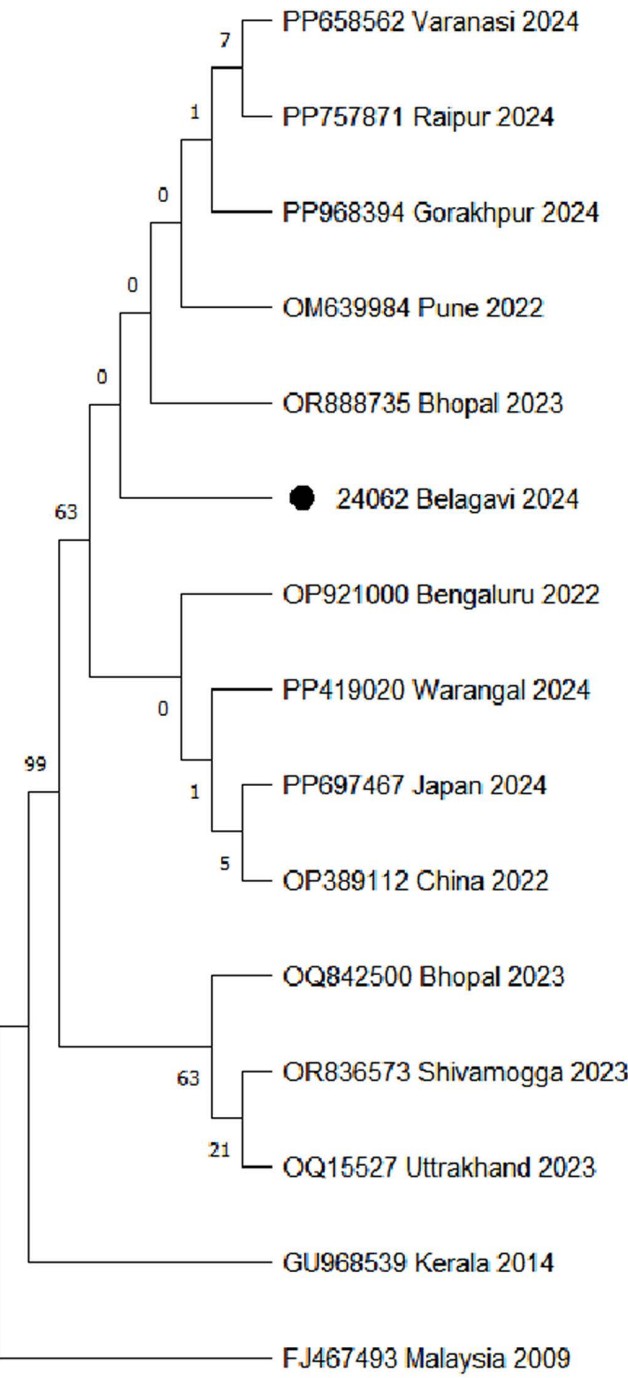

**Fig 2. Evolutionary analysis by the Maximum Likelihood method\*.** \*Phylogenetic analysis was performed using the Maximum Likelihood method and the Tamura-Nei model in MEGA 11 software [13,14]. The tree incorporated 15 aligned nucleotide sequences (456 positions after pairwise deletion), with branch support evaluated using 1,000 bootstrap replicates. The Belagavi 2024 isolate (24062 Belagavi 2024) clustered closely with the Bhopal 2023 (OR888735) strain (bootstrap value=63), indicating recent shared ancestry.

replication continued into the seroconversion phase, indicating active disease transmission during this period. The detection of both biomarkers in the same patient suggests that these individuals may have presented to healthcare facilities during the transition phase of the disease, between the acute and convalescent phases [9,10,17].

Haematological findings varied significantly by serological status: NS1-positive subjects had lower leukocyte and platelet counts, consistent with early-phase infection; while IgM-positive subjects, likely in later stages, showed more pronounced thrombocytopenia and elevated transaminases, indicating hepatic injury—hallmarks of immune-mediated damage [20]. Remarkably, creatinine levels were significantly higher among patients with warning signs, suggesting early renal involvement in those at risk for severe disease. Kidney dysfunction in dengue is increasingly recognized and mandates close monitoring [20,21].

DENV-2 infections were associated with a broader clinical presentation, including systemic and gastrointestinal symptoms, which is consistent with existing literature that links DENV-2 to more symptomatic and clinically diverse infections. In contrast, the patients infected with DENV-1 and DENV-3 exhibited a limited symptom profile, confined to fever and myalgia, although the small sample size limits generalizability. Overall, the clinical features such as fever and myalgia were nearly universal, consistent across serotypes as per previous studies [22]. The absence of severe complications across all serotypes suggests that the observed patients predominantly represented the milder end of the dengue disease spectrum. It may also be a reflection of effective clinical management or the predominance of primary, rather than secondary infections.

The high prevalence of DENV-2 (91.30%) among the tested samples is consistent with previous studies conducted in different parts of India, where DENV-2 has been identified as the dominant circulating serotype during outbreaks [5,22]. DENV-1 and DENV-3 were detected in only a few patients (4.76% each), with no patients of DENV-4 or DENV-5 found during this study. This result aligns with the observation that while multiple serotypes are known to co-circulate, DENV-2 remains the most prevalent in many endemic regions of India [9]. With regards to studies from Karnataka, the earlier study from the Mysuru district revealed that DENV-4 was the predominant serotype in 2016.[23] Followed by a resurgence of DENV-1 and DENV-2 serotypes in 2017, reported by Prasanna N Y et al. and a recent study by Shrutthi U et al, reported a predominance of DENV-3 in eight zones of urban Bangalore among the patients reported from January to July 2024 [24,25] and contrasting a predominance of DENV-2 from a study targeting the rural areas of Bangalore [26] as shown in **Fig 3**.

Despite our best efforts, the high-quality sequence was obtained from only 1/7 RT-PCR-positive samples. The low positivity may be a reflection of declining viral loads below the threshold level for amplification, which may, in turn, be because the majority of patients were in the convalescent or post-viraemic phase. The single sequence provides a preliminary molecular snapshot which confirms the circulating genotype/lineage as DENV-2 and offers a potential indication of its regional relatedness to an isolated Bhopal strain. This pattern is consistent with prior molecular surveillance studies demonstrating limited intra-serotype diversity and region-specific clustering of DENV-2 in India [22,27–29]. However, the potential for future genetic drift or recombination events necessitates continued molecular surveillance to detect emerging variants with altered virulence, immune evasion, or transmission potential.

## 5.1 Regional context and public health implications

Karnataka has witnessed a surge in dengue patients, with 9,880 patients reported from January to July 2024, compared to 2,271 patients in the same period in 2023, highlighting the significant disease burden in the state. Our study provides the first comprehensive molecular surveillance data from North Karnataka (Belagavi), filling an important gap in understanding regional dengue epidemiology. The predominance of a single serotype (DENV-2) during the 2024 season suggests that the population may be at risk for secondary infections with other serotypes in future transmission cycles. Notably, DENV-2 was the predominant serotype (91.3%), reflecting recent trends in Karnataka [12,15]. Given serotype shifts and the risk

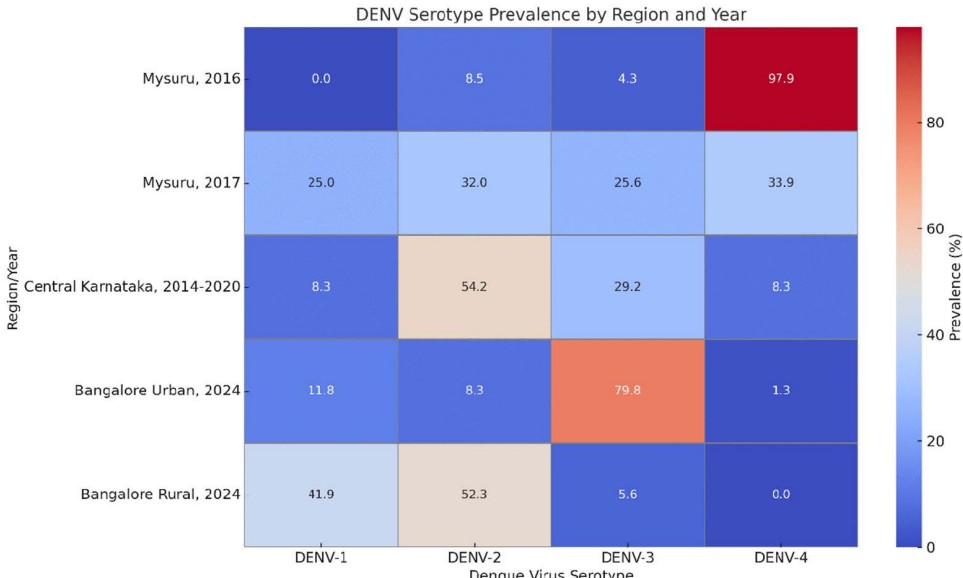

**Fig 3. Heatmap infographic shows the distribution of DENV serotypes across different regions of Karnataka (2016-2024) \*.** *The colour intensity reflects the proportion of each serotype, making it easy to visualise the dominance of specific serotypes in each region.

of antibody-dependent enhancement in secondary infections (especially with DENV-2), sustained molecular surveillance remains critical for outbreak anticipation and guiding vaccine policy [6,22].

## 6. Limitation

This study had several limitations. The present study findings serve as a sentinel surveillance report providing the first molecular characterization from the region, not a statistically representative cross-section of all cases. The small sample size for RT-PCR confirmation and genetic sequencing reduced statistical power and regional representativeness. Most patients were enrolled during the late or convalescent phases, resulting in low NS1 antigen detection and limited viral RNA recovery for serotype identification. Only one case yielded high-quality genomic sequence data for phylogenetic analysis, restricting insights into the genetic diversity of circulating dengue strains. IgG results of the Bioline Dengue Duo RDT were not consistently validated by an accompanying IgG capture ELISA in the VRDL lab and were therefore excluded to ensure data quality and focus on the primary study objective. We acknowledge the lack of IgG data for the classification of primary vs. secondary infections.

## 7. Conclusion

This study provides the first molecular surveillance of DENV serotypes in Belagavi, Karnataka, during 2024, revealing DENV-2 predominance (91.30%). And also serves as a critical, first-ever local snapshot of the circulating serotype (DENV-2), which is valuable for public health surveillance in the region and emphasizes the need for a sustained molecular surveillance infrastructure. High late-phase diagnoses indicate critical gaps in early detection and healthcare-seeking behaviours, necessitating improved diagnostic strategies and community awareness. Future surveillance should prioritise early sample collection, enhanced diagnostic sensitivity, and real-time data sharing for improved outbreak detection. Integrating molecular surveillance with clinical outcomes and vector data will support effective public health interventions and evidence-based dengue control strategies in the region.

## Supporting information

**S1 Table. Serotype-specific laboratory data of complete blood count, liver function test, and renal function test and serotype distribution across thrombocytopenia grades.**
(DOCX)

**S2 Table. Table depicting the estimates of evolutionary divergence between sequences analysed.**
(DOCX)

**S3 Table. GenBank Sequences utilized for phylogenetic analysis.**
(DOCX)

**S4 Table. Raw data.**
(XLSX)

## Acknowledgments

Special thanks are extended to the VRDL project staff, Ms. Aishwarya S, Ms. Shabnam C, Dr. Manjunath R C, Ms. Ankita Saha, Mr. Sandeep Panchal, Mr. Gautam N and Mr. Sunil K., for their assistance and collaboration throughout the study. The authors also express heartfelt gratitude to Dr. Kalesh M Karun, Scientist C, and Mr. Flemin Felix, Project Technical Support II, both from ICMR-NITM Belagavi, for their expert guidance and support in the biostatistical analysis of the data. **Declaration of generative AI and AI-assisted technologies in the writing process:** During the preparation of this work, the author(s) used Claude AI and ChatGPT to improve readability, language and create Figures. After using this tool, the author(s) reviewed and edited the content as needed and take(s) full responsibility for the content of the publication.

## Author contributions

**Conceptualization:** Rekha Patil, Madhav Prabhu, Kamran Zaman, Jyothi Bhat.

**Data curation:** Rekha Patil, Asif Kavathekar, Madhav Prabhu, Madhavi Patil, Ameya Kejriwal, Kamran Zaman, Jyothi Bhat.

**Formal analysis:** Asif Kavathekar, Madhavi Patil, Ameya Kejriwal, Kamran Zaman, Jyothi Bhat.

**Funding acquisition:** Kamran Zaman.

**Investigation:** Asif Kavathekar, Madhavi Patil, Ameya Kejriwal, Kamran Zaman, Jyothi Bhat.

**Methodology:** Rekha Patil, Asif Kavathekar, Madhav Prabhu, Kamran Zaman, Jyothi Bhat.

**Project administration:** Kamran Zaman, Jyothi Bhat.

**Supervision:** Kamran Zaman, Jyothi Bhat, Subarna Roy.

**Validation:** Rekha Patil, Asif Kavathekar, Madhav Prabhu, Madhavi Patil, Ameya Kejriwal.

**Writing – original draft:** Asif Kavathekar, Madhavi Patil, Ameya Kejriwal.

**Writing – review & editing:** Rekha Patil, Madhav Prabhu, Kamran Zaman, Jyothi Bhat, Subarna Roy.

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
