## [Decision Letter · Decision Letter 0]

28 Oct 2025

Tracking the Clinico-microbiological profile and molecular evolution of Dengue cases during the Monsoon-Season in Belagavi, Karnataka

Dear Dr. Bhat,

Thank you for submitting your manuscript to PLOS Neglected Tropical Diseases. After careful consideration, we feel that it has merit but does not fully meet PLOS Neglected Tropical Diseases's publication criteria as it currently stands. Therefore, we invite you to submit a revised version of the manuscript that addresses the points raised during the review process.

Please submit your revised manuscript within 60 days Dec 27 2025 11:59PM. If you will need more time than this to complete your revisions, please reply to this message or contact the journal office at plosntds@plos.org. Please include the following items when submitting your revised manuscript:

We look forward to receiving your revised manuscript.

Kind regards,

Husain Poonawala

Academic Editor

Elvina Viennet

Section Editor

Shaden Kamhawi

co-Editor-in-Chief

Paul Brindley

co-Editor-in-Chief

**Additional Editor Comments :**

The paper does not have a clear hypothesis and the results do not support the data presented

The article needs careful copyediting and proof-reading. For example, Table 1 states Igm, when it should be IgM

Please clarify how 89 patients were chosen from the 794 patients with dengue for further analysis. Why were only 28 samples testing for serotyping?

Were there any considerations about false-positivity due to lower specificity of the IgM?

I would not include interpretation of results in the results - please use those in the discussion. For example, "suggesting higher vulnerability among young adults" is subjective and debatable. Young adults maybe at higher risk because they are of working age and hence might have a higher exposure.

If the SD Bio line Dengue Duo assay tested IgG and IgM, why is IgG not reported? For section 3.2, it is unclear which results are from rapid testing and which are from ELISA - I would make that clearer.

IF Figure 1 is supposed to be a bubble plot, please include bubbles and not crosses

Unclear what the relevance of section 3.3 and 3.4. the sample size is not large enough to make conclusions that associate clinical features with a specific serotype.

I would not make any conclusions about molecular epidemiology with only 1 sequenced sample, especially since the authors were able to obtain high-quality data with only 1 of 23 samples with RT-PCR.

The discussion includes multiple speculative statements that are not supported by the results.

**Journal Requirements:**

At this stage, the following Authors/Authors require contributions: Rekha Patil, Asif Kavathekar, Madhav Prabhu, Madhavi Patil, Ameya Kejriwal, Kamran Zaman, Jyothi Bhat, and Subarna Roy. Please ensure that the full contributions of each author are acknowledged in the "Add/Edit/Remove Authors" section of our submission form.

4) We have noticed that you cited Supplementary Figure 2 on page 8 in your manuscript. However, there is no corresponding file uploaded to the submission. Please upload it as a separate file with the item type 'Supporting Information'.

5) We have noticed that you have uploaded Supporting Information files, but you have not included a list of legends. Please add a full list of legends for your Supporting Information files after the references list.

6) When completing the data availability statement of the submission form, you indicated that you will make your data available on acceptance. We strongly recommend all authors decide on a data sharing plan before acceptance, as the process can be lengthy and hold up publication timelines. Please note that, though access restrictions are acceptable now, your entire data will need to be made freely accessible if your manuscript is accepted for publication. This policy applies to all data except where public deposition would breach compliance with the protocol approved by your research ethics board. If you are unable to adhere to our open data policy, please kindly revise your statement to explain your reasoning and we will seek the editor's input on an exemption. Please be assured that, once you have provided your new statement, the assessment of your exemption will not hold up the peer review process.

7) Please amend your detailed Financial Disclosure statement. This is published with the article. It must therefore be completed in full sentences and contain the exact wording you wish to be published.

8) Please revise your current Competing Interest statement to the standard "The authors have declared that no competing interests exist."

8) Please ensure that the funders and grant numbers match between the Financial Disclosure field and the Funding Information tab in your submission form. Note that the funders must be provided in the same order in both places as well. Currently, "the Indian Council of Medical Research (ICMR)" is missing from the Funding Information tab.

**Comments to the Authors:**

**Please note that one review is uploaded as an attachment.**

**Reviewers' Comments:**

Reviewer's Responses to Questions

**Key Review Criteria Required for Acceptance?**

**Methods**

-Are the objectives of the study clearly articulated with a clear testable hypothesis stated?

-Is the study design appropriate to address the stated objectives?

-Is the population clearly described and appropriate for the hypothesis being tested?

-Is the sample size sufficient to ensure adequate power to address the hypothesis being tested?

-Were correct statistical analysis used to support conclusions?

-Are there concerns about ethical or regulatory requirements being met?

Reviewer #1: The study claims to be the first comprehensive molecular surveillance report from northern part of Karnataka and highlights the predominance of DENV-2 serotype during the study period. The current study has clearly stated objectives & hypothesis as noted in the article. The study design is appropriate to address the stated objectives.

It is a prospective clinico-microbiological research study undertaken jointly by a hospital and a research facility with clearly stated study design, appropriate population as noted and clearly defined methods involving clinical examination, laboratory techniques including ELISA, RT-PCR and sequencing analysis. The sample size and the statistical tools applied for analysis appear appropriate for the conclusions derived. The authors have shared details of the ethical clearance obtained for undertaking the study on clinical subjects. 89 dengue-confirmed cases out of the 794 dengue-positive cases detected using NS1 antigen and IgM antibody by ELISA technique were subjected to serotyping by RT-PCR targeting the CprM gene & a subset of the positive RT-PCR specimens was subjected to sequencing and phylogenetic analysis in the study.

The authors have also highlighted the limitations of the study, including the sample size. Similar studies undertaken in the future will help in understanding the changing trends and patterns of clinical presentation of Dengue cases and also the serotypes associated with the condition.

Reviewer #2: Why were only 28 NS1-positive cases tested by RT-PCR? Could testing additional RNA-extractable samples have given a more information regarding the circulating serotypes?

2. Out of 794 dengue-positive cases, only 28 NS1-positive samples were tested by RT-PCR. Are these few samples really representative of all the cases? Would testing more NS1-positive (or possibly IgM-positive) samples have given stronger and more reliable conclusions about the circulating serotypes?

3. Sequencing Strategy:- Only a subset of RT-PCR positive samples underwent sequencing. Please provide details on how samples were chosen for sequencing and whether this may have biased phylogenetic interpretation.

**Results**

-Does the analysis presented match the analysis plan?

-Are the results clearly and completely presented?

-Are the figures (Tables, Images) of sufficient quality for clarity?

Reviewer #1: The results portrayed in the study are in line with the defined objectives. The analysis presented by the authors match with the analysis plan. The results and supporting details in the form of tables and figures appear to be appropriate. Supplementary data submitted by the authors has also been reviewed and found to be satisfactory.

Reviewer #2: 4. Can the findings from this single tertiary care centre be generalized to the wider population of northern Karnataka?”

4. Clinical Correlations:- The manuscript reports symptom associations with serotypes (e.g., chills, retro-orbital pain with DENV-2). Given the small number of non–DENV-2 cases, how reliable are these associations?

5. What specific factors determined which NS1-positive samples were chosen for further analysis?

6. NS1 antigen is typically detectable only during the first 5 days of illness. After this period, NS1 results may turn negative even in confirmed dengue cases. Do the authors have data on the number of days between symptom onset and sample collection? If available, please include this information to support the explanation for low NS1 positivity

7. Hematological and Biochemical Parameters:- Thrombocytopenia and creatinine levels are mentioned briefly but not fully discussed. Could the authors elaborate on the relevance of hematological and biochemical parameters, and indicate whether any kidney involvement was observed in severe cases?

8. For the samples that were tested for dengue serotypes, can the authors provide the corresponding CBC, RFT, and LFT results? Presenting this data in a table in the supplementary material would help readers see any clinical differences between serotypes.

**Conclusions**

-Are the conclusions supported by the data presented?

-Are the limitations of analysis clearly described?

-Do the authors discuss how these data can be helpful to advance our understanding of the topic under study?

-Is public health relevance addressed?

Reviewer #1: The conclusions drawn are supported by the data presented by the authors and establishes a reference for future similar studies in understanding the clinical trends and changing serotype distribution of Dengue viruses. The authors have described the limitations of the analysis and have also undertaken to share the relevant details of the study, if required.

Another notable feature of the study also lies in emphasizing the importance of continuous molecular surveillance to track viral evolution and monitor serotype shift and genetic variations.

Reviewer #2: This study provides important molecular data from northern Karnataka, a region with limited previous reports. It combines clinical, serological, and molecular findings, shows that DENV-2 is the dominant serotype, and highlights implications for diagnosis, surveillance, and vaccine planning.

**Editorial and Data Presentation Modifications?**

Reviewer #1: The content of the article seems appropriate for the journal. I herewith express my recommendation 'ACCEPT. The editorial team may consider alignment and use of high resolution images as deemed appropriate for the final script.

Reviewer #2: (No Response)

**Summary and General Comments**

Reviewer #1: The novelty lies in the fact that it is the first of its kind study endeavored to provide insights into the characterise circulating DENV serotypes and correlate them with the clinical presentation of patients in the geographical region. The study undertaken provides useful insights into the clinical presentation of Dengue cases and association of the various serotypes associated with the infection. The authors have undertaken the study with clearly defined objectives, a suitable study design intending to provide tangible results using appropriate data analysis. The high proportion of late-phase diagnoses of dengue reported in the study stresses on the requirement of improved early detection strategies and timely interventions to manage dengue outbreaks, especially during high-risk transmission periods. A larger sample size may be incorporated in future studies for determining the clinical association and also to map the serotypes prevalent in the geographical region. Constant molecular surveillance to track viral evolution and monitor serotype shift and genetic variations is recommended for epidemiological as well as clinical correlation. The results of the study would also be of immense potential in planning the preventive & treatment strategies by the health authorities.

Reviewer #2: This manuscript entitled “Tackling the clinic-microbiological profile and molecular evaluation of Dengue cases during Monsoon-season in Balagavi, Karnataka” presents a prospective, hospital-based observational study conducted in Belagavi, Karnataka, during the 2024 monsoon season. The authors examined 89 confirmed dengue patients out of 794 dengue-positive cases. Serotyping was performed on NS1-positive samples, revealing DENV-2 as the predominant serotype (91.3%), followed by DENV-1 and DENV-3. Phylogenetic analysis of DENV-2 strains indicated close relationships with isolates from Bhopal, suggesting possible regional transmission dynamics. Clinically, fever and myalgia were predominant, while DENV-2 cases more often presented with chills, headache, and retro-orbital pain. The study highlights late-phase diagnosis (high IgM positivity), raising concerns about delays in detection, and emphasizes the importance of continued molecular surveillance in guiding outbreak preparedness and vaccine strategies.

PLOS authors have the option to publish the peer review history of their article (what does this mean? ). If published, this will include your full peer review and any attached files.

**Do you want your identity to be public for this peer review?** For information about this choice, including consent withdrawal, please see our Privacy Policy .

Reviewer #1: No

Reviewer #2: **Yes:** 0a

**Figure resubmission:**
---

## [Decision Letter · Decision Letter 1]

19 Dec 2025

Dear Dr Bhat,

We are pleased to inform you that your manuscript 'Tracking the clinico-microbiological profile and molecular characterization of dengue cases during the monsoon-season in Belagavi, Karnataka' has been provisionally accepted for publication in PLOS Neglected Tropical Diseases.

Best regards,

Husain Poonawala

Academic Editor

David Safronetz

Section Editor

Shaden Kamhawi

co-Editor-in-Chief

Paul Brindley

co-Editor-in-Chief

Reviewer's Responses to Questions

**Key Review Criteria Required for Acceptance?**

**Methods**

-Are the objectives of the study clearly articulated with a clear testable hypothesis stated?

-Is the study design appropriate to address the stated objectives?

-Is the population clearly described and appropriate for the hypothesis being tested?

-Is the sample size sufficient to ensure adequate power to address the hypothesis being tested?

-Were correct statistical analysis used to support conclusions?

-Are there concerns about ethical or regulatory requirements being met?

Reviewer #1: The study claims to be the first comprehensive molecular surveillance report from northern part of Karnataka and highlights the predominance of DENV-2 serotype during the study period. The current study has clearly stated objectives & hypothesis as noted in the article. The study design is appropriate to address the stated objectives.

The authors have revised and resubmitted the manuscript. They have now clearly stated that the study's primary objectives as an essential descriptive molecular and clinico-epidemiological surveillance effort for an area (North Karnataka) where such data is critically absent. The revised Introduction and Abstract now clearly state two main aims. They also intend to use the more appropriate term 'Molecular characterization' in place of 'Molecular evolution' in the title of the manuscript.

It is a prospective clinico-microbiological research study undertaken jointly by a hospital and a research facility with clearly stated study design, appropriate population as noted and clearly defined methods involving clinical examination, laboratory techniques including ELISA, RT-PCR and sequencing analysis. The sample size and the statistical tools applied for analysis appear appropriate for the conclusions derived. The authors have shared details of the ethical clearance obtained for undertaking the study on clinical subjects. 89 consecutive subset of dengue-confirmed cases out of the 794 dengue-positive cases detected using NS1 antigen and IgM antibody by ELISA technique were subjected to serotyping by RT-PCR targeting the CprM gene & a subset of the positive RT-PCR specimens was subjected to sequencing and phylogenetic analysis in the study.

The authors have also highlighted the limitations of the study, including the sample size. Similar studies undertaken in the future will help in understanding the changing trends and patterns of clinical presentation of Dengue cases and also the serotypes associated with the condition.

Reviewer #2: Revised and accepted

**Results**

-Does the analysis presented match the analysis plan?

-Are the results clearly and completely presented?

-Are the figures (Tables, Images) of sufficient quality for clarity?

Reviewer #1: The results portrayed in the revised manuscript are in line with the defined objectives. The analysis presented by the authors match with the analysis plan. The results and supporting details presented in the revised manuscript in the form of tables and figures appear to be appropriate. Supplementary data submitted by the authors has also been reviewed and found to be satisfactory. 89 consecutive subset of dengue-confirmed cases out of the 794 dengue-positive cases detected using NS1 antigen and IgM antibody by ELISA technique were subjected to serotyping by RT-PCR targeting the CprM gene & a subset of the positive RT-PCR specimens was subjected to sequencing and phylogenetic analysis in the study.

The authors have reasoned that of the 23 RT-PCR positive samples, only one yielded high quality sequence data, which is a limitation of the study. This low yield has been attributed to the low concentration and fragmentation of viral RNA due to the late presentation of patients, a common challenge for molecular work in resource-limited settings.

Reviewer #2: revised and accepted

**Conclusions**

-Are the conclusions supported by the data presented?

-Are the limitations of analysis clearly described?

-Do the authors discuss how these data can be helpful to advance our understanding of the topic under study?

-Is public health relevance addressed?

Reviewer #1: The conclusions drawn are supported by the data presented by the authors and establishes a reference for future similar studies in understanding the clinical trends and changing serotype distribution of Dengue viruses. The authors have described the limitations of the analysis and have also undertaken to share the relevant details of the study, if required.

Another notable feature of the study also lies in emphasizing the importance of continuous molecular surveillance to track viral evolution and monitor serotype shift and genetic variations.

The conclusions presented in the revised manuscript are restricted to the evidence presented on the predominant serotype (DENV-2), the clinical profile of the N=89 cohort, and the high proportion of late-phase diagnoses.

Reviewer #2: revised and accepted

**Editorial and Data Presentation Modifications?**

Reviewer #1: The content of the article seems appropriate for the journal. I herewith express my recommendation 'ACCEPT. The editorial team may consider alignment and use of high resolution images as deemed appropriate for the final script.

Reviewer #2: revised and accepted

**Summary and General Comments**

Reviewer #1: The revised manuscript has used the more appropriate term 'molecular characterization' in place of 'molecular evolution'. The novelty lies in the fact that it is the first of its kind study endeavored to provide insights into the characterize circulating DENV serotypes and correlate them with the clinical presentation of patients in the geographical region. The study undertaken provides useful insights into the clinical presentation of Dengue cases and association of the various serotypes associated with the infection. 89 consecutive subset of dengue-confirmed cases out of the 794 dengue-positive cases detected using NS1 antigen and IgM antibody by ELISA technique were subjected to serotyping by RT-PCR targeting the CprM gene & a subset of the positive RT-PCR specimens was subjected to sequencing and phylogenetic analysis in the study.

The authors have undertaken the study & have revised the manuscript with clearly defined objectives, a suitable study design intending to provide tangible results using appropriate data analysis. The conclusions presented in the revised manuscript are restricted to the evidence presented on the predominant serotype (DENV-2), the clinical profile of the N=89 cohort, and the high proportion of late-phase diagnoses. The authors have mentioned that of the 23 RT-PCR positive samples, only one yielded high quality sequence data, which is one of the limitations of the study. The available sequence data would be insufficient to draw conclusions at a community or regional level. The high proportion of late-phase diagnoses of dengue reported in the study stresses on the requirement of improved early detection strategies and timely interventions to manage dengue outbreaks, especially during high-risk transmission periods. A larger sample size may be incorporated in future studies for determining the clinical association and also to map the serotypes prevalent in the geographical region. Constant molecular surveillance to track viral evolution and monitor serotype shift and genetic variations is recommended for epidemiological as well as clinical correlation. The results of the study would also be of immense potential in planning the preventive & treatment strategies by the health authorities.

Reviewer #2: revised and accepted

PLOS authors have the option to publish the peer review history of their article (what does this mean? ). If published, this will include your full peer review and any attached files.

**Do you want your identity to be public for this peer review?** For information about this choice, including consent withdrawal, please see our Privacy Policy .

Reviewer #1: No

Reviewer #2: **Yes:** Ravisekhar Gadepalli

---

## [Editor Report · Acceptance letter]

Dear Dr Bhat,

We are delighted to inform you that your manuscript, "

Tracking the clinico-microbiological profile and molecular characterization of dengue cases during the monsoon-season in Belagavi, Karnataka," has been formally accepted for publication in PLOS Neglected Tropical Diseases.

Best regards,

Shaden Kamhawi

co-Editor-in-Chief

Paul Brindley

co-Editor-in-Chief
